# MIRO1 Is Required for Dynamic Increases in Mitochondria-ER Contact Sites and Mitochondrial ATP During the Cell Cycle

**DOI:** 10.3390/cells14070482

**Published:** 2025-03-22

**Authors:** Benney T. Endoni, Olha M. Koval, Chantal Allamargot, Tara Kortlever, Lan Qian, Riley J. Sadoski, Denise Juhr, Isabella M. Grumbach

**Affiliations:** 1Abboud Cardiovascular Research Center, Division of Cardiovascular Medicine, Department of Internal Medicine, Carver College of Medicine, University of Iowa, Iowa City, IA 52242, USA; benney-endoni@uiowa.edu (B.T.E.); olha-koval@uiowa.edu (O.M.K.); tara-kortlever@uiowa.edu (T.K.); lan-qian@uiowa.edu (L.Q.); riley-sadoski@uiowa.edu (R.J.S.); denise-juhr@uiowa.edu (D.J.); 2Interdisciplinary Graduate Program in Molecular Medicine, University of Iowa, Iowa City, IA 52242, USA; 3Central Microscopy Research Facility, University of Iowa, Iowa City, IA 52242, USA; chantal-allamargot@uiowa.edu; 4Iowa City VA Healthcare System, Iowa City, IA 52246, USA

**Keywords:** mitochondria, ER, MERCS, MAM, MIRO1, cell cycle, vascular smooth muscle cells, fibroblasts, Ca^2+^

## Abstract

Mitochondria-ER contact sites (MERCS) are vital for mitochondrial dynamics, lipid exchange, Ca^2+^ homeostasis, and energy metabolism. We examined whether mitochondrial metabolism changes during the cell cycle depend on MERCS dynamics and are regulated by the outer mitochondrial protein mitochondrial rho GTPase 1 (MIRO1). Wound healing was assessed in mice with fibroblast-specific deletion of MIRO1. Wild-type and MIRO1^-/-^ fibroblasts and vascular smooth muscle cells were evaluated for proliferation, cell cycle progression, number of MERCS, distance, and protein composition throughout the cell cycle. Restoration of MIRO1 mutants was used to test the role of MIRO1 domains; Ca^2+^ transients and mitochondrial metabolism were evaluated using biochemical, immunodetection, and fluorescence techniques. MERCS increased in number during G1/S compared with during G0, which was accompanied by a notable rise in protein–protein interactions involving VDAC1 and IP3R as well as GRP75 and MIRO1 by proximity-ligation assays. Split-GFP ER/mitochondrial contacts of 40 nm also increased. Mitochondrial Ca^2+^ concentration ([Ca^2+^]), membrane potential, and ATP levels correlated with the formation of MERCS during the cell cycle. MIRO1 deficiency blocked G1/S progression and the cell-cycle-dependent formation of MERCS and altered ER Ca^2+^ release and mitochondrial Ca^2+^ uptake. MIRO1 mutants lacking the Ca^2+^-sensitive EF hands or the transmembrane domain did not rescue cell proliferation or the formation of MERCS. MIRO1 controls an increase in the number of MERCS during cell cycle progression and increases mitochondrial [Ca^2+^], driving metabolic activity and proliferation through its EF hands.

## 1. Introduction

Mitochondria-ER contact sites (MERCS) are specialized regions where the membranes of the mitochondria and the endoplasmic reticulum (ER) come into close proximity, facilitating direct interaction between the two organelles [1,2,3,4]. Comprised of several protein complexes that bridge the organelles, MERCS play a key role in processes such as phospholipid synthesis [5], bioenergetics, and Ca^2+^ signaling [6]. Voltage-dependent anion channels (VDACs) on the outer mitochondrial membrane (OMM) form a complex with inositol-1,4,5-trisphosphate receptors (IP3Rs) and glucose-related protein 75 (GRP75), enabling rapid Ca^2+^ transfer [2,7,8]. Through the close alignment of mitochondria and the ER at MERCS, Ca^2+^ released from intracellular stores, primarily the ER, is taken up by mitochondria via the mitochondrial Ca^2+^ uniporter (MCU) [9,10]. The influx of Ca^2+^ activates Ca^2+^-dependent hydrogenases that drive NADH and, ultimately, ATP production, thereby meeting the energy demands of the cell [11]. In proliferating cells, Ca^2+^ influx changes with the cell cycle, peaking during the G1/S phase and beyond [12,13]. Nevertheless, the mechanisms that modulate mitochondrial Ca^2+^ during the cell cycle remain under investigation [14,15,16]. Given the central role of MERCS in Ca^2+^ dynamics, and a recent report showing that these contacts expand during mitosis [17], we investigated if the number of MERCS or their protein complex composition regulate Ca^2+^ transfer during cell cycle progression.

Mitochondrial Ca^2+^ homeostasis and motility are among several intracellular processes that require the outer mitochondrial protein mitochondrial rho GTPase 1 (MIRO1) [18,19]. MIRO1 attaches to the outer mitochondrial membrane via a transmembrane domain and has two EF hands flanked by two GTPase domains [20]. MIRO1 facilitates intracellular mitochondrial distribution and motility via Trak, microtubules, and MYO [19], and regulates mitochondrial transport via its Ca^2+^-sensitive EF-hand domains; when Ca^2+^ binds to MIRO1’s EF hands, it induces a conformational change that triggers mitochondrial arrest and dissociation from microtubules. Beyond its role in mitochondrial motility, MIRO1 also associates with proteins within MERCS [21,22,23,24,25] and modulates mitochondrial Ca^2+^ influx [25,26]. In several cell types including neurons and fibroblasts, MIRO1 deficiency is implicated in the dysfunction of IP3R-GRP75-VDAC1 MERCS [21,25], but it is unknown whether the MERCS/MIRO1 interaction controls eukaryotic cellular processes, such as proliferation.

This study investigated whether MERCS are dynamically regulated with the cell cycle in proliferating cells such as vascular smooth muscle cells (VSMCs) and fibroblasts, and whether MIRO1 influences MERC dynamics and mitochondrial metabolic activity during proliferation. Specifically, MIRO1 deletion was analyzed for effects on the formation of MERCS, the composition of protein complexes at contact sites, Ca^2+^ distribution, and mitochondrial metabolic activity throughout the cell cycle. Additionally, MIRO1 mutants of the functional domains, i.e., the Ca^2+^-sensitive EF hands, C-terminal GTPase, and transmembrane domain, were tested for their capacity to regulate MERCS. Finally, a mouse model with a fibroblast-specific MIRO1 deletion was utilized to assess the effect of MIRO1 on wound healing in vivo.

## 2. Materials and Methods

Animals

Animal studies were conducted in accordance with the NIH Guide for the Care and Use of Laboratory Animals, following protocols approved by the Institutional Animal Care and Use Committee (IACUC) of the University of Iowa. The mitochondrial rho1 GTPase LoxP (floxed Miro1 conditional KO) mice were graciously provided by Dr. Janet Shaw (University of Utah). The fibroblast-specific Cre transgenic mice, Col1a2CreERT (*Col1a2* cis-acting fibroblast-specific enhancer with minimal promoter; Jackson Laboratory strain# 029567), were a kind gift from Dr. Long-Sheng Song (University of Iowa). The floxed Miro1 conditional knockout mice were crossed with Col1a2CreERT mice to generate tamoxifen-inducible, fibroblast-specific Miro1 knockouts. Cre recombination was induced by intraperitoneal injection of tamoxifen (80 mg/kg; Sigma, Cat# T5648) for 5 days followed by a 7-day break. Tamoxifen injections were then resumed for an additional 5 days. MIRO1^fl/fl^ mice were injected with the same dose of tamoxifen and used as a wild-type (WT) control for the in vivo experiments.

Skin wounding

Mice were anesthetized with isoflurane, and the skin on their dorsal side was shaved the day before the wounding assay. Mice were 11–16 weeks old at the time of wounding. Male and female mice were used in equal proportions. All control and study group mice underwent skin wounding. To create full-thickness skin wounds, mice were anesthetized, the wound site was cleaned with a surgical scrub, and a sterile 3.5 mm biopsy punch (Miltex, Cat# 33–34) was used to excise a sample of the dorso-rostral back skin and panniculus carnosum. Care was taken not to injure underlying muscles. Digital photographs were acquired on the day of surgery and every day thereafter for up to 7 days. The photographer was unaware of the genotype of the mice. The wound area was calculated using ImageJ software 1.54f to define margins. No mice were excluded from the analysis.

Cell isolation and culture

Skin fibroblasts were isolated from male and female mice by enzymatic dispersion. The tissue was minced and digested overnight in a solution of Dulbecco’s Modified Eagle’s Medium (DMEM) without fetal bovine serum (FBS) and with fungizone (0.01%; ThermoFisher, Cat# 15290-018), penicillin/streptomycin (1%; GIBCO, Cat# 15140-122), and collagenase I (1 mg/mL; Worthington Biochemical Corporation, Cat# LS004194) at 37 °C in the presence of 5% CO_2_. The following day, the digested tissue was gently agitated by pipetting to dissociate cells and centrifuged at 1500× *g* for 5 min. The pelleted cells were suspended in DMEM supplemented with 10% FBS (Hyclone, Cat# SH30910.03), 1% penicillin/streptomycin, MEM non-essential amino acids (GIBCO, Cat# 15630-080), MEM Vitamin Solution (Sigma/Cat# M6895), and 8 mmol/L HEPES (GIBCO, Cat# 15630-080) and cultured at 37 °C in humidified air with 5% CO_2_. Cells from passages 3 to 7 were used.

Primary aortic VSMCs were isolated from Miro1^fl/fl^ mouse aortas by enzymatic digestion. Isolated aortas were incubated in collagenase type II (1 mg/mL; Worthington Biochemical Corporation, Cat# LS004202) at 37 °C for 10 min to remove the adventitia. The medial layer of the aorta, which contains the VSMCs, was minced and incubated in collagenase type II (2 mg/mL) for 3 h at 37 °C. Cells were suspended in DMEM supplemented with 20% FBS, 1% penicillin/streptomycin, MEM non-essential amino acids, MEM Vitamin Solution, and 8 mmol/L HEPES and cultured at 37 °C in humidified air with 5% CO_2_. Cells from passages 3 to 9 were used.

MIRO1 deletion was generated by transducing cultured Miro1^fl/fl^ fibroblasts or VSMCs with an adenovirus expressing Cre recombinase at a multiplicity of infection (MOI) of 50 for 48 h. Deletion of MIRO1 was confirmed by immunoblotting. The transduction was repeated if immunoblots showed that MIRO1 expression was >50% of WT levels.

Cell counting

Skin fibroblasts were cultured in 12-well plates (Corning, Cat# 3513) with 10,000 cells per well in 2 mL DMEM containing 10% FBS. Twenty-four hours after seeding, cells were treated with or without PDGF (20 ng/mL; Sigma, Cat# SRP3229) and cultured for an additional 72 h. Cells were then trypsinized with Gibco™ 0.25% Trypsin-EDTA (Cat# 25200-056) and counted in triplicate using a Beckman Coulter Z1 cell counter.

Cell cycle synchronization

Serum starvation was used to synchronize the cell cycle of VSMCs and skin fibroblasts. Cells were incubated in FBS-free DMEM supplemented with 1% penicillin/streptomycin, MEM non-essential amino acids, MEM Vitamin Solution and 8 mmol/L HEPES for 48 and 72 h, respectively. Serum-starved cells were used as the 0 h samples. Cells were then released from growth arrest by incubation with 10% FBS for 24 or 48 h.

Flow-cytometry analysis of cell cycle progression

Skin fibroblasts were synchronized and released from growth arrest as described above. A total of 1 × 10^6^ cells were harvested and fixed with 70% ethanol for 1 h at 4 °C. Cells were then centrifuged at 850× *g* for 5 min and the pellets rinsed with PBS and stained with propidium iodide solution (100 µg/mL). Cytometry was performed using the Becton Dickinson LSR II instrument (560 nm laser). Forward scatter (FS) and side scatter (SS) were measured to identify single cells, and pulse processing (PI area vs. PI width) was used to exclude cell doublets from the analysis. The data were acquired using BD FACS Diva software 8.0.1.

Construction and transduction of MIRO1 cDNA-expressing adenoviruses

The MIRO1 plasmid constructs pRK5-myc-Miro1 (# 47888), pRK5-myc-Miro1 E208K/E328K (# $7894), pRK5-myc-Miro1 S432N (# 47893), and pRK5-myc-Miro1 Δ593–618 (# 47895) were obtained from Addgene (Watertown, MA, USA). The Miro1 gene full-length ORF (GenBank ID: AJ517412) of each construct was subcloned into the adenoviral shuttle vector pacAd5-CMV-mcs SV40pA (University of Iowa Viral Vector Core) by incorporating 5′ ClaI and 3′ EcoRI restriction sites. Positive clones were identified by restriction digestion.

Skin fibroblasts were transduced with adenovirus constructs at a multiplicity of infection (MOI) of 50 for 48 h. Expression of WT or MIRO1 mutants in MIRO1^−/−^ cells was confirmed by immunoblotting.

Construction and transduction of SPLICS_L_ and nucleofection of SPLICS_S_ cDNA

The SPLICS Mt-ER long P2A (# 164107) plasmid was purchased from Addgene. The full-length ORF of the P2A peptide was subcloned into the NotI (5′) and EcoRI (3′) restriction sites of the adenoviral shuttle vector pacAd5-CMV-mcs SV40pA (Agilent) from the University of Iowa Viral Vector Core. Positive clones were identified by restriction digestion. VSMCs or skin fibroblasts were transduced with the adenovirus construct at a multiplicity of infection (MOI) of 50 for 48 h.

The SPLICS Mt-ER short P2A (# 164108) plasmid was purchased from Addgene. VSMCs were electroporated with the SPLICS_S_ plasmid in a Nucleofector^TM^ I device (Lonza) with the Basic Nucleofector^TM^ Kit for Primary Mammalian Smooth Muscle Cells (Lonza # VPI-1004), following the manufacturer’s protocol. A total of 50,000 cells were electroporated in the presence of 5 μg plasmid DNA, plated onto 35 mm glass-bottom microwell dishes (MatTek Corporation, Ashland, MA, USA, Cat# P35G-1.5-20-C) and cultured for 48 h before experiments were performed.

SPLICS_S_ and SPLICS_L_ puncta were quantified using an ImageJ plugin as described by Giamogante et al. [27].

Confocal microscopy

For live-cell imaging, VSMCs cells were plated onto 35 mm MatTek dishes pre-coated with 0.1% gelatin and transduced with SPLICS_L_ adenovirus overnight or electroporated with SPLICS_S_ before plating. After serum starvation or release from growth arrest, mitochondria were labeled with MitoTracker Deep Red (LifeTechnologies Corporation, Carlsbad, CA, USA, Cat# M22425) in HBSS (GIBCO, Cat# 14025-092) at 37 °C for 30 min. Cells were then washed with HBSS, and images were acquired using an LSM 510 confocal microscope at a magnification of 63× (Carl Zeiss, San Diego, CA, USA). All images were taken at the same time using the same imaging settings. Image analysis was performed using the SPLICS quantification plugin in ImageJ.

Transmission electron microscopy and analysis of the mitochondrial ultrastructure

VSMCs were washed with 1× Dulbecco’s PBS (DPBS), fixed with 2.5% glutaraldehyde in 0.1 M sodium cacodylate buffer (EM grade, pH 7.4) overnight at 4 °C, and postfixed with 1% osmium tetroxide for one hr. The samples were dehydrated by submersion in a series of ethanol concentrations (50% to 100%) and then embedded in Epon 812 (Ted Pella, Redding, CA, USA), which was incubated at 70 °C overnight to polymerize. Samples were sectioned using a Leica EM UC7 ultramicrotome. The ultrathin sections (50 nm) were collected on formvar/carbon-coated copper grids and then stained with 2.5% uranyl acetate and lead citrate for 2 min, respectively. Stained sections were screened with a JEOL JEM-1230 transmission electron microscope (JEOL, Peabody, MA) and photographed using a Gatan UltraScan 1000 2k × 2k Tietz CCD camera (Gatan, Pleasanton, CA, USA) at the Central Microscopy Research Facility at the University of Iowa.

Duolink Proximity Ligation Assay (PLA)

VSMCs were plated onto 35 mm glass-bottom dishes with 10 mm bottom wells (Cellvis, Cat# D35-10-1.5-N) pre-coated with 0.1% gelatin. Cells were washed with PBS, fixed with 4% PFA at 37 °C, and stored at 4 °C until all the cells from all the time points were collected. The cells were permeabilized in PBS with 0.1% Tween, followed by blocking with Duolink^®^ Blocking solution (Sigma Aldrich, (St. Louis, MO, USA) Cat# DUO82007) for 1 h at 37 °C and overnight incubation with primary antibodies at 4 °C. A proximity-ligation assay was performed using Duolink^®^ In Situ Red PLA reagents (Cat# DUO92004 and DUO92002) according to the manufacturer’s protocol. Images were captured using a Nikon Eclipse Ti2 inverted microscope. PLA signals in in situ images were counted using ImageJ by applying the “Analyze Particles” function following threshold correction.

MAM isolation

A total of 5 × 10^6^ skin fibroblasts was used for each MAM isolation. A Percoll gradient was used to separate MAMs from other intracellular fractions by ultracentrifugation at 100,000× *g* for 45 min. Subcellular fractionation was performed as previously described [28].

Co-immunoprecipitation

HEK cells were transduced with adenovirus encoding c-Myc tagged MIRO1, MIRO1 KK, MIRO1 dnC, or MIRO1 ΔTM for 24 h at an MOI of 2. Whole-cell lysates were collected. Protein concentrations were measured using the Pierce™ BCA protein assay (Thermo Scientific (Waltham, MA, USA), Cat# 23227), and 2 mg of lysates were used. c-Myc mouse antibody (0.5 μg/reaction; Santa Cruz, sc-40) was precipitated using magnetic Dynabeads Protein G (25 μL/reaction; Thermofisher Scientific 10003D) following a 30 min incubation at room temperature. Specifically, the beads were washed 3 times with Pierce washing buffer (25 mM Tris HCl, 150 mM NaCl, 1% NP-40, 1 mM EDTA, 5% glycerol) and incubated with protein lysates (1 mg) overnight at 4 °C. After three washes, protein was eluted from magnetic beads by boiling in lithium dodecyl sulfate (LDS) for 10 min at 70 °C. Samples were then run on 4–12% acrylamide, Bis-Tris gels for immunoblotting.

Transfection with MIRO1 siRNAs

HEK cells were seeded in 10 cm cell-culture plates in DMEM medium supplemented with 10% FBS, 1% penicillin/streptomycin, MEM non-essential amino acids, MEM Vitamin Solution, and 8 mmol/L HEPES at 37 °C in a humidified 95% air and 5% CO_2_ incubator. The siRNA duplexes targeting MIRO1 were 5′-GUUGUUGCAGAUAUCUCAGAAUCG-3′ and 5′-UCAGUGUCAGCUUACCAACAUGACA-3′ (IDT hs. RiRHOT1.13.1 and 3) or a scrambled negative control (5′-CUUCCUCUCUUUCUCUCCCUUGUGA-3′). Transfection was performed using Lipofectamine 2000 (Invitrogen Cat# 11668027). A total of 5 nM of scrambled or MIRO1 siRNAs, 5 µL of RNAiMAX, and 500 μl of Gibco Opti-MEM (Thermofisher Scientific, Cat# 31985-070) were incubated together at room temperature for 30 min. DMEM media was replaced with 5 mL of Opti-MEM, and the siRNA cocktail was added to the cells. Transfection proceeded for 3 h, and then 5 mL of DMEM media was added. After 48 h, the knockdown efficiency was evaluated by immunoblotting.

Cell lysis and fractionation

Cells were lysed in RIPA buffer (20 mM Tris, 150 mM NaCl, 5 mM EDTA, 5 mM EGTA, 1% Triton X-100, 0.5% deoxycholate, 0.1% SDS, pH 7.4) supplemented with both a protease inhibitor cocktail (Complete Mini, Cat# 11836153001 Roche) and phosphatase inhibitors (PhosSTOP, Cat# 04906837001 Roche). Lysates were sonicated, and the debris was pelleted by centrifugation at 10,000× *g* for 10 min at 4 °C. Mitochondrial fractions were prepared in MSEM buffer (5 mM MOPS, 70 mM sucrose, 2 mM EGTA, 220 mM Mannitol, pH 7.5 with protease inhibitors), with homogenization performed in ice-cold MSEM buffer in a Potter–Elvehjem glass Teflon homogenizer (50 strokes). Nuclei and cell debris were pelleted by centrifugation at 600× *g* for 5 min at 4 °C. Mitochondria were separated from the cytosolic fraction by centrifugation at 8000 × g for 10 min at 4 °C. Protein concentrations were determined using the Pierce™ BCA protein assay (Thermo Scientific, Cat# 23227).

Immunoblotting

All protein samples were lysed by homogenization in RIPA buffer supplemented with phosphatase inhibitors and protease inhibitors. Samples were sonicated briefly and centrifuged at 5000× *g* for 10 min. Protein concentrations in the supernatants were quantified using the Pierce BCA protein assay. Equal amounts of protein (10–50 µg) were resolved via PAGE on Novex 4–12% SDS NuPAGE precast gels (Life Technology, Cat# NP0335BOX). Gels were transferred to Immobilon-P 0.45 mm PVDF membranes. Membranes were incubated with the primary antibody in 5% BSA in 0.05% Tween-20 in TBS (TBS-T) buffer at 4 °C overnight. Blots were washed 3 times for 5 min with TBS-T and incubated for 60 min at room temperature with secondary antibodies at a dilution of 1:5000. Blots were developed with an ECL chemiluminescent substrate using the iBright 1500 imager (Invitrogen, Waltham, MA, USA) according to the manufacturer’s instructions and quantitated using the Image Lab 6.0 software (Bio-Rad, Hercules, CA, USA). Densitometry was performed using ImageJ software.

Measurement of ER Ca^2+^ transients

The CEPIA1*er* plasmid was purchased from Addgene (#58218). VSMCs were transfected in a Nucleofector I device (Lonza, Basel, Switzerland) using the Basic Nucleofector Kit for Primary Mammalian Smooth Muscle Cells (VPI-1004, Lonza) and following the manufacturer’s protocol. A total of 600,000 cells was electroporated with 5 μg of plasmid DNA, plated onto 35 mm glass-bottom microwell dishes (MatTek Corporation), and grown for 48 h before synchronization and release from growth arrest. Before imaging, the media was replaced with HBSS. Imaging was performed at room temperature using a Nikon Eclipse Ti2 inverted microscope. The CEPIA1*er* protein was excited at 543 nm, and the emitted fluorescence was measured at a 580 nm wavelength. PDGF (20 ng/mL) was added by micropipette to trigger ER Ca^2+^ release. Recordings were measured every 5 s for 5 min and quantified using GraphPad Prism 10.0 software. The peak amplitude over baseline and the area under the curve (AUC) after PDGF application were calculated. The peak amplitude was determined by subtracting the change in fluorescence from the baseline signal, after adding the agonist.

HEK cells were also transfected with 5 μg of the CEPIA1*er* plasmid DNA using Lipofectamine 2000 for 48 h. Before imaging, the media was replaced with FluoroBrite DMEM (Invitrogen, Cat# A1896701). ATP (1 mM; Research Product International, Mt Prospect, IL, USA, Cat# A300030) was added by micropipette to trigger ER Ca^2+^ release.

Measurement of cytosolic Ca^2+^ transients

VSMCs were loaded with 2 µM Fura 2-acetoxymethyl ester (Fura 2-AM, Invitrogen Cat# F1201) by incubation in HBSS for 20 min at room temperature. Cells were washed twice with HBSS and incubated at 37 °C for 5 min for de-esterification. Cells were excited alternatively at 340 and 380 nm. The fluorescence signal was acquired at 510 nm. Real-time shifts in the Fura-2AM fluorescence ratio were recorded after the addition of PDGF (20 ng/mL) every 5 s for 10 min using a Nikon Eclipse Ti2 inverted microscope. Recordings were analyzed using GraphPad Prism. The peak amplitude was calculated by subtracting the baseline fluorescence ratio from the peak fluorescence ratio. The area under the curve (AUC) was normalized by subtracting the AUC at baseline. Summary data represent the average differences in the basal and peak increases in [Ca^2+^]_cyto_.

Thapsigargin 1 µM (Sigma-Aldrich, TG # T9033), SEA 0400 1 μM (NCX inhibitor; Cayman, #19876), and Xestospongin C 5 μM (IP3R inhibitor; Cayman, # 64950) were added to the medium for 1 h before Fura 2-AM loading.

HEK cells were incubated in FluoroBrite DMEM with 1 µM Fura 2-AM for 20 min at room temperature. Before imaging, the media was replaced with FluoroBrite and ATP (100 µM) was added by micropipette.

Mitochondrial Ca^2+^ imaging

Ratiometric Ca^2+^ measurements in the mitochondria were performed using adenovirus-delivered mtPericam, a fluorescent Ca^2+^-indicator protein with a COX VIII targeting sequence. VSMCs were transduced with Ad-mtPericam at an MOI of 50 at 24 h prior to serum starvation. Ratiometric fluorescent imaging of mtPericam via a Nikon Eclipse Ti2 inverted microscope was used to determine the intensity of the fluorescence signal, with excitation at 415 nm and 480 nm and emission at 510 nm. Baseline [Ca^2+^] was recorded every 5 s for 15 s. For Ca^2+^ transients, PDGF (20 ng/mL) was added by micropipette to trigger mitochondrial Ca^2+^ uptake, and recordings were measured every 5 s for 10 min. The peak amplitude and AUC were determined as described for Fura 2-AM. Time-to-peak was analyzed using the Boltzmann equation in Graphpad Prism for fitting the rising (ascending) part of PDGF-induced Ca^2+^-transients:Y=Bottom+(Top−Bottom)1+exp(t1/2−Xslope)
where slope is the rate of change in curve amplitude and is inversely related to the time-to-peak, i.e., a smaller slope yields a steeper curve and a shorter time-to-peak, while a larger slope yields a shallow curve and a longer time-to-peak. The parameter t_1/2_ represents the time value that is halfway between the minimum and maximum curve plateau.

For MERC-independent mitochondrial Ca^2+^ uptake, cells were permeabilized with media supplemented with digitonin 0.001% (Millipore-Sigma, 300410) 2 h prior to each trace. Recordings were performed in Ca^2+^-free Hanks Buffer. Ca^2+^ boluses (CaCl_2_) were added at 5.0 µM, and the responses were recorded for 2 min. Peak amplitude was normalized to baseline for quantification.

As an additional method, HEK cells were transfected with the CEPIA4*mt* plasmid DNA purchased from Addgene (# 58220). HEK cells were transfected for 48 h with 5 μg of plasmid DNA using Lipofectamine 2000. Before imaging, the media was replaced with FluoroBrite DMEM (Invitrogen, Cat# A1896701). ATP (100 μM) was added by micropipette to trigger mitochondrial Ca^2+^ uptake.

Mitochondrial membrane potential

Mitochondrial membrane potential was quantified by measuring the fluorescence of tetramethylrhodamine methyl ester (TMRM; Life Technology, T668). Cells were incubated with TMRM (20 nM) for 15 min at 37 °C. The staining solution was then rinsed off and replaced with HBSS. The TMRM fluorescence was excited at 528 nm and emitted at 580 nm and recorded using a Nikon Eclipse Ti2 inverted microscope.

Mitochondrial ATP measurement

Mitochondrial ATP levels were assessed with Biotracker ATP Red Live Cell Dye (SCT 045; Millipore, Sigma-Aldrich). Cells were transduced with mitoGFP adenovirus at an MOI of 50 during synchronization or incubated with 100 nM MitoTracker green (LifeTechnologies Corporation, Cat# M7514) for 15 min, and then incubated with 10 μM of ATP Red dye for 5 min. Cells were washed once with HBSS and imaged using a Nikon Eclipse Ti2 inverted microscope. The RFP:GFP ratio was used as an indicator of mitochondrial ATP levels. For control experiments, cells were treated with oligomycin (1 µM, 10 µM, 100 µM) overnight or FCCP (10 µM) for 30 min. WT cells synchronized at 0 h then released from growth arrest for 24 h and at 48 h were treated overnight with Ru265 (100 µM), then incubated with 10 µM of ATP Red dye for 5 min, and the RFP:GFP ratio was measured.

Intracellular ATP measurement

The ratiometric, cytoplasmic-ATP-sensor plasmid was purchased from Addgene (# 102551). The fluorescent ATP-sensitive protein pm-iATPSnFR1.0 was subcloned into the adenoviral shuttle vector pacAd5-CMV-mcs SV40pA using EcoRI (5′) and XhoI (3′) restriction sites. Adenoviral particles were generated by the University of Iowa Viral Vector Core. Cells were transduced with the adenovirus (MOI of 50) for 24 h during synchronization. Fluorescence was measured on a Nikon Eclipse Ti2 inverted microscope. Cells were excited at 486 and 558 nm. The GFP signal intensity was measured at 510 nm and that of the mRuby signal was measured at 605 nm. The GFP:mRuby ratio was used as an indicator of cytosolic ATP levels.

Statistical analysis

Data were analyzed using GraphPad Prism and expressed as the mean ± SEM. Normality and equal variance were assessed. Statistical comparisons between two groups were carried out using the unpaired *t*-test or the Mann–Whitney U-test when a normal distribution could not be assumed. In experiments in which two groups were compared and the test sample was normalized to the WT control set as 1, the Wilcoxon matched-pairs signed rank test was used. One-way analysis of variance (ANOVA), followed by appropriate comparison tests, was used for multiple group comparisons. Two-way ANOVA, followed by appropriate post hoc tests, was used for grouped data sets. A *p*-value of <0.05 was considered significant. In the figures, *p*-values are indicated for comparisons between time points in one genotype or treatment and for comparisons between treatments at the same time point.

## 3. Results

### 3.1. MIRO1 Is Required for Fibroblast Proliferation and Cell Cycle Progression

To examine whether MIRO1 affects cell proliferation, transgenic MIRO1^fl/fl^ mice were crossbred with fibroblast-specific, tamoxifen-inducible Cre transgenic mice (Col1a2-CreERT) (Figure 1A) and treated with tamoxifen to selectively delete MIRO1 from fibroblasts. MIRO1 deficiency was confirmed by immunoblots of cultured skin fibroblast extracts (Figure 1B). To induce proliferation of skin fibroblasts explanted from WT and Miro1^−/−^ mice, platelet-derived growth factor (PDGF) was added to the culture media. Compared with untreated cells, PDGF treatment for 72 h induced robust proliferation in WT fibroblasts, while fibroblasts from PDGF-treated Miro-1^−/−^ mice proliferated far less (Figure 1C). To identify the stage of the cell cycle affected by MIRO1 deletion, skin fibroblasts were synchronized at the G0/G1 phase (termed 0 h) by a 72 h period of serum starvation and then released from growth arrest with media containing 10% FBS for 24 and 48 h (Appendix A). In contrast with MIRO1^-/-^ fibroblasts, WT fibroblasts expressed increased levels of cyclin D1 and E, indicators of the G1/S phase, 24 h after release from growth arrest (Appendix A). These results were validated by FACS analysis where, at 24 h after release from growth arrest, about 50% of the WT fibroblasts had entered the S phase compared with less than 20% of MIRO1^−/−^ cells (Figure 1D,E). Together, these data demonstrate that MIRO1 deficiency prevents fibroblast proliferation by impairing the G1/S phase of the cell cycle. MIRO1 deficiency was also tested in an in vivo assay of wound healing of a cutaneous punch biopsy. Compared with WT mice, at day 3, wound closure was significantly delayed in Miro1^−/−^ mice (Figure 1F,G).

### 3.2. MIRO1 Deficiency Abolishes Cell Cycle Changes at MERCS

Next, cell cycle changes in the ER–mitochondria distance were assessed by measuring the number or cleft distances of mitochondrial ER contact sites (MERCS). As previously reported, split-GFP-based contact sensors (SPLICS) can be expressed [29], with SPLICS_L_ and SPLICS_S_ indicating a cleft distance of approximately 40 nm and 10 nm, respectively. Equal expression of the SPLICS in WT and MIRO1^−/−^ cells was verified by immunoblotting for GFP (Appendix A). In cycling WT VSMCs, the number of SPLICS_L_ fluorescence signals increased significantly at 24 h compared with 0 h, while SPLICS_S_ signals were highest at 0 h and decreased at 24 and 48 h, suggesting that the interaction between mitochondria and the ER changes throughout the cell cycle. In MIRO1^−/−^ VSMCs, however, neither SPLICS_S_ nor SPLICS_L_ signals significantly changed during the cell cycle (Figure 2A–D), and significantly fewer SPLICS_L_ signals were detected at 24 h in MIRO1^-/-^ compared with WT VSMCs. Additionally, the ER–mitochondrial distance in proliferating, non-synchronized VSMCs from WT and smooth-muscle-cell-specific MIRO1 knockout mice was analyzed by transmission electron microscopy (Qian, in review) [30]. Overall, the distance was reduced with MIRO1 deficiency compared with WT conditions (Appendix A).

### 3.3. MIRO1 Associates with MERCS During the Cell Cycle

Cell cycle changes in interactions between proteins within MERCS were monitored via an in situ proximity ligation assay (PLA). Specifically, the interactions tested were between IP3R and VDAC1, the organelle-surface proteins involved in the Ca^2+^ transfer complex at the MAM interface, and between GRP75 and MIRO1, as well as between VAPB and PTPIP51, which are proteins involved in lipid transfer. At 24 h, compared with at 0 h, WT cells showed increased interactions between IP3R and VDAC1 (Figure 2E,F; Appendix A), and GRP75 and MIRO1 (Figure 2G,H; Appendix A), but these interactions did not increase in MIRO1^−/−^ cells. Interestingly, in both WT and MIRO1^−/−^ cells, VAPB-PTPIP51 interactions were highest at 0 h compared with at 24 and 48 h and were not different between genotypes (Appendix A). These data suggest that MIRO1 selectively regulates the formation of the MERCS involved in Ca^2+^ transfer.

### 3.4. MIRO1 Abundance Increases in MAMs in G1/S

MIRO1 regulates cell cycle-dependent changes in MERC protein–protein interactions. Therefore, MIRO1 abundance at the mitochondria-associated membrane (MAM) interface might also fluctuate and impact other proteins throughout the cell cycle. To test this, MAMs were isolated, and the quality of the preparation was confirmed by immunoblotting for markers of the various subcellular fractions (Appendix A). MAM expression of IP3R, VDAC1, GRP75, as well as MIRO1 was assayed in WT fibroblasts at 0 h vs. 24 h, which showed that MIRO1 localized to both the mitochondrial and MAM fractions, with a significant increase in the MAM fraction at 24 h (Figure 3A,B). In addition, while IP3R, GRP75, and VDAC1 were also detected in the MAM fraction (Figure 3A,C,D), their abundance did not consistently change during the cell cycle. Moreover, the abundance of VAPB and FACL4 increased at 24 h, particularly in WT cells (Figure 3A,E,F).

### 3.5. MIRO1 EF Hands Are Required for Its Association with GRP75 and MCU

To identify the MIRO1 functional domain that regulates the formation of MERCS, MIRO1 protein–protein interactions were monitored in co-immunoprecipitation (co-IP) assays in HEK cells, comparing the proteins that interact with WT vs. MIRO1 mutants, i.e., mutants lacking the EF-hand domains (MIRO1 KK), the dominant-negative C-terminal GTPase domain (MIRO1 dnC), or the transmembrane domain (MIRO1 ΔTM). These co-immunoprecipitations detected an association between MIRO1 and both GRP75 and MCU, but not with IP3R, VDAC1, FACL4, or VAPB (Figure 3G). Furthermore, associations were reduced with MIRO1 KK and, in particular, MIRO1 ΔTM mutants; in contrast, MIRO1 dnC associated with GRP75 and MCU at levels comparable to that of the WT (Figure 3H–J). Taken together, these findings suggest that MIRO1 is a resident of MAMs and physically interacts with proteins involved in Ca^2+^ transfer, and that its Ca^2+^-dependent EF hands as well as its anchoring in the outer mitochondrial membrane are required for the association.

The effect of short-term MIRO1 deficiency on other MAM proteins was also evaluated. In VSMCs, MIRO1 deletion caused increased expression of IP3R (Appendix A). Moreover, in HEK cells with acute siRNA silencing of MIRO1, the expression levels of GRP75, IP3R, and VDAC1 proteins were significantly increased (Appendix A).

### 3.6. MIRO1 Promotes ER–Mitochondria Ca^2+^ Transfer During Cell Cycle Progression

The IP3R-GRP75-VDAC1 complex is a critical mediator of Ca^2+^ signals from the ER to mitochondria [7,31]. Previous studies demonstrated that subcellular [Ca^2+^] fluctuate with the cell cycle [32,33]. Therefore, proliferating WT vs. MIRO1^−/−^ VSMCs were compared for differences in [Ca^2+^]. PDGF-induced ER Ca^2+^ transients measured with CEPIA1*er* were highest at 0 h, decreased somewhat at 24 h, and significantly decreased at 48 h. Notably, in MIRO1^−/−^ cells, ER Ca^2+^ release was reduced at 0 h compared with WT controls and remained unchanged at later time points (Figure 4A,B; Appendix A). To measure the effect on cytosolic [Ca^2+^], Fura 2-AM recordings were performed following the addition of PDGF. In WT cells after PDGF treatment, the peak amplitude and AUC were highest at 0 h and decreased at 24 and 48 h but remained unchanged in MIRO1^−/−^ cells (Figure 4C,D; Appendix A).

To investigate how ER Ca^2+^ store depletion impacts cytosolic [Ca^2+^], we performed additional recordings using thapsigargin, a reagent that inhibits the sarco/endoplasmic reticulum Ca^2+^-ATPase (SERCA) (Appendix A). Following the addition of thapsigargin, cytosolic [Ca^2+^] levels in WT cells showed a consistent pattern, with higher levels observed at 0 h compared with those at 48 h. MIRO1^-/-^ cells exhibited consistently lower and unchanged cytosolic [Ca^2+^] across different time points (Appendix A).

To analyze the effects of IP3R and NCX on cytosolic [Ca^2+^] in WT versus MIRO1^-/-^ VSMCs, a series of Fura 2-AM recordings was performed using PDGF stimulation following pretreatment with either the NCX inhibitor (SEA 0400) or the IP3R inhibitor xestospongin C (Appendix A). The analysis revealed no significant differences in cytosolic [Ca^2+^] between WT and MIRO1^-/-^ cells, indicating that MIRO1 deletion does not impact IP3R or NCX function. These findings further suggest that ER [Ca^2+^] levels are highest at 0 h, coinciding with the lowest number of contact sites in WT VSMCs. Conversely, fluctuations in both MERCS and Ca^2+^ levels are significantly reduced in MIRO1^−/−^ cells over time. Since ER [Ca^2+^] depletion can induce ER stress, ER-stress-protein expression was tracked by immunoblotting; in WT and MIRO1^-/-^ VSMCs, however, the levels of the ER stress markers XBP1 and CHOP were similar (Appendix A).

Next, mitochondrial Ca^2+^ uptake and [Ca^2+^] were measured using the mitochondrial Ca^2+^ indicator, mtPericam. In WT cells, mitochondrial Ca^2+^ transients were lowest at 0 h and increased significantly by 48 h (Figure 4E,F; Appendix A). Additionally, the time-to-peak in WT cells shortened as the cell cycle progressed, indicating that a steeper curve with a shorter time-to-peak contributes to the higher peak amplitude. Notably, this pattern was absent in MIRO1^−/−^ VSMCs (Appendix A). Moreover, the baseline [Ca^2+^]_mito_ was elevated at 24 h (Figure 4G). In MIRO1^-/-^ cells, however, PDGF-induced transients and baseline [Ca^2+^]_mito_ did not significantly change during the cell cycle.

Immunoblots of WT and MIRO1^−/−^ VSMCs demonstrated no differences in the expression of MCUcomplex proteins (i.e., MCU and MICU1; Appendix A), suggesting that this abolished increase in mitochondrial Ca^2+^ uptake and the low basal [Ca^2+^]_mito_ in MIRO1^−/−^ cells are not due to MCUcomplex deficiency. Next, MERCs-independent mitochondrial [Ca^2+^] uptake was tested in permeabilized WT and MIRO1^−/−^ cells by directly adding 5 µM [Ca^2+^] (Appendix A). The results showed no differences between the genotypes at any time point in the cell cycle. Additionally, the mitochondrial membrane potential was measured using TMRM, showing a significant increase at 24 h in WT cells and a comparatively lower membrane potential in MIRO1^−/−^ cells (Figure 4H). These data reveal that the distribution of subcellular [Ca^2+^] fluctuates with the cell cycle in WT but not MIRO^−/−^ VSMCs.

Lastly, to test whether MIRO1 regulates Ca^2+^ transfer independently of the cell cycle, ER, mitochondrial, and cytosolic Ca^2+^ transients were measured using CEPIA1*er*, CEPIA4mt, and Fura 2-AM, respectively, in unsynchronized HEK cells. Acute knockdown of MIRO1 by siRNA significantly increased ATP-induced ER Ca^2+^ transients, in particular the peak amplitude (Appendix A), while cytosolic Ca^2+^ levels remained unaltered (Appendix A). In MIRO1 knockdown cells, mitochondrial Ca^2+^ entry was significantly decreased in both the peak amplitude and AUC (Appendix A).

### 3.7. MIRO1 Is Required to Maintain Intracellular ATP Levels During Cell Cycle Progression

Mitochondrial Ca^2+^ regulates the activity of pyruvate dehydrogenase (PDH) by promoting its dephosphorylation and activation [11,34], a post-translational modification that can be tracked via immunoblotting. In WT VSMCs, 24 h after cell cycle re-entry, PDH Ser293 was mostly dephosphorylated. In contrast, PDH in MIRO1^−/−^ cells displayed a high baseline phosphorylation that did not further decrease (Figure 5A,B). Since the activity of TCA cycle dehydrogenases impacts ATP synthesis, mitochondrial ATP levels were measured using a fluorescent ATP-Red live-cell dye. The specificity of the dye was verified in cells treated with oligomycin, a mitochondrial ATP synthase inhibitor, or FCCP, which uncouples mitochondrial oxidative phosphorylation (Appendix A); ATP-Red fluorescence was normalized to mitoGFP fluorescence (Appendix A). With this approach, mitochondrial ATP levels were significantly increased in WT VSMCs as well as in fibroblasts at 24 h (Figure 5C, Appendix A). Blocking mitochondrial Ca^2+^ influx by pharmacological inhibition of MCU with Ru265 abolished this effect (Appendix A).

Furthermore, mitochondrial ATP contributes to intracellular ATP levels. Thus, a decrease in mitochondrial ATP in MIRO1^-/-^ cells could affect intracellular ATP levels. Accordingly, at 24 h compared with 0 h, WT cytosolic ATP levels in both VSMCs and fibroblasts were increased but remained unchanged in MIRO1^-/-^ cells (Figure 5D,E; Appendix A). Taken together, these data demonstrate that MIRO1 supports mitochondrial Ca^2+^ influx, PDH activation, and the mitochondrial ATP production.

### 3.8. MIRO1 EF-Hand Domains Are Required for MERCS Formation and Proliferation

Since the EF hands and the C-terminal mitochondrial transmembrane domain were required for the maximal association of MIRO1 with other MERCS proteins (Figure 3G), we hypothesized that the mutants MIRO1 KK, which lacks the EF hands, and MIRO1 ΔTM, which lacks the mitochondrial transmembrane domain, would not be able to fully rescue MIRO1^−/−^ defects in MERCS formation, ATP levels, and cell proliferation. To confirm this expectation, MIRO1 WT, MIRO1 KK, MIRO1 dnC, or MIRO1 ΔTM was expressed in MIRO1^−/−^ skin fibroblasts (Appendix A), and SPLICS_L_ signals were monitored. At 24 h, SPLICS_L_ signals were as abundant in MIRO1^−/−^ fibroblasts expressing MIRO1 WT or MIRO1 dnC as in WT cells but not in MIRO1^−/−^ fibroblasts expressing MIRO1 KK or MIRO1 ΔTM (Figure 6A,B). Similarly, MIRO1 WT or MIRO1 dnC-expressing MIRO1^−/−^ fibroblasts demonstrated significantly higher mitochondrial ATP levels at 24 h compared with those expressing MIRO1 KK or MIRO1 ΔTM (Figure 6C). Likewise, PDGF treatment induced significant cell proliferation in MIRO1^−/−^ fibroblasts after reconstitution with MIRO1 WT or MIRO1 dnC but not with MIRO1 KK or MIRO1 ΔTM (Figure 6D,E). These data demonstrate that the Ca^2+^-sensitive EF hands and transmembrane domain of MIRO1 are required for MERCS formation and mitochondrial ATP production and drive VSMC proliferation.

## 4. Discussion

Mitochondria communicate with ER at MERCS, where their membranes are closely aligned. MERCS can adapt by changing the length of the mitochondrial–ER apposition, the gap distance, and the number of close contact sites under cellular stress [29]. This study demonstrates that the protein MIRO1 is crucial for regulating these changes during the cell cycle in the G1/S phase. The loss of MIRO1 disrupts the number, gap distance, and composition of MERCS. MIRO1 must be anchored in the outer mitochondrial membrane and have functional EF hands to influence the IP3R/VDAC1 contact sites, which are essential for Ca^2+^ transfer in proliferating cells.

Recent studies on MERCS in cell proliferation have utilized visualization methods such as split-GFP constructs or inducible linkers [17,35]. These tools revealed an increase in contact sites during late cell cycle stages and mitosis and increased mitochondrial [Ca^2+^]. However, they did not define the gap distance or focus on metabolite transfer to mitochondria. Our study contributes to this body of evidence by detecting changes in the number and lengths of contact sites at an earlier stage than previously reported—in the G1/S phase. Furthermore, we provide parallel evidence of increased Ca^2+^ transfer and ATP levels in this phase, which is known to require high ATP production levels. Our data also implicate MIRO1 as a crucial regulator. We identify that this GTPase drives these effects through its association with GRP75, requiring Ca^2+^-dependent EF hands.

Previous studies highlighted the role of MIRO1 in directing mitochondrial transport along microtubules [36,37]. We also reported that MIRO1-dependent mitochondrial mobility is necessary for cell migration [38]. Here, we established that the loss of MIRO1 is sufficient to delay cell cycle progression at the G1/S phase and to halt fibroblast proliferation and wound healing. As the underlying mechanism, we propose that changes in MERCS abundance and distances during the cell cycle impact on metabolic activity and are regulated by MIRO1.

While several studies have implicated MIRO1 in cell proliferation, none have defined a role for the in vivo phenotypes of pathological and physiological proliferation. Recent work from our lab has shown an impairment in VSMC proliferation in vitro and reduced neointima formation in vivo (Qian, submitted) [30]. A previous study reported on MIRO1’s role in wound healing in C. elegans, albeit through mitochondrial fragmentation [39]. In embryonic fibroblasts, the deletion of both MIRO1 and MIRO2 impaired the symmetric microtubule-dependent segregation of mitochondria in mitosis and mitosis rates [40]. Moreover, the loss of both MIRO1 and MIRO2 was necessary to decrease contacts between the ER and mitochondria, which correlated with delayed mitochondrial Ca^2+^ uptake [25].

Our findings indicate that MERCS are highly dynamic structures that undergo remodeling during the cell cycle. In the G1/S phase, the number of 40 nm wide contact sites increased significantly, while those spanning 8–10 nm decreased. Deletion of MIRO1 blocked the dynamic increase in 40 nm wide contact sites. While the functional relevance of different distances of contact sites requires further studies, recent findings indicate that the optimal distance for Ca^2+^ flux between the ER and mitochondria is 20 nm, which is longer than previously postulated [41,42]. In our hands, reconstitution with MIRO1 KK, which lacks the EF hands, did not rescue these changes, supporting the idea that MIRO1, and its EF hands in particular, drive these cell-cycle-associated dynamics.

Additionally, we present the novel finding that MIRO1 is necessary for changes in the composition of MERCS, which consists of the IP3R-GRP75-VDAC1 complex. The colocalization of IP3R-VDAC1 and GRP75-MIRO1 increased in G1/S in WT cells but not in cells with MIRO1 deletion. This suggests that MIRO1 regulates the interaction underlying [Ca^2+^]^+^ transport at MERCS. In addition, the pulldown assay corroborates these findings: MIRO1 physically interacted with GRP75 and MCU, which was weaker in the presence of MIRO1 KK. Moreover, in G1/S, MAMs contained significantly more MIRO1 protein. Notably, protein–protein interactions between VAPB and PTPIP51 in MERCS were lower in G1/S compared with G0. The MERC proteins have been implicated in Ca^2+^ as well as phospholipid transfer [43,44,45]. MIRO1 deletion had no effects on these changes, providing strong evidence that these aspects of MERCS are not regulated by MIRO1. Taken together, these findings imply that MERCS formation is highly regulated during the cell cycle, and that MIRO1 is critical only for regulating MERCS containing the IP3R-GRP75-VDAC1 complex.

These studies confirm that MIRO1 is required to boost mitochondrial ATP production by activating Ca^2+^-mediated dehydrogenase in the mitochondrial matrix. This extends previous work from our lab showing that MIRO1 controls ATP levels, electron-transport-chain activity, and mitochondrial cristae formation [30]. MIRO1 promotes mitochondrial Ca^2+^ entry, which boosts mitochondrial ATP production by driving the activity of TCA-cycle dehydrogenases, including pyruvate dehydrogenase (PDH). During the G1/S phase, basal mitochondrial [Ca^2+^] and agonist-induced Ca^2+^ transients increase in WT conditions but not with MIRO1 deletion. Based on our studies of ER, cytosolic, and mitochondrial [Ca^2+^], we posit that the differences in mitochondrial and cytosolic Ca^2+^ transients at 0 h versus later time points in WT cells are predominantly driven by variations in ER Ca^2+^ loading that negatively correlate with the density of MERCS. Accordingly, MIRO1 boosts PDH dephosphorylation/activation and mitochondrial ATP levels, and both are blunted with MIRO1 deletion. Previous studies suggest that mitochondrial ATP output increases to meet the metabolic demands of G1/S progression, and that mitochondrial Ca^2+^ uptake plays a major role [46]. These new data confirm that MIRO1 is important in that process and shows that, mechanistically, MIRO1 regulates the Ca^2+^ entry needed for mitochondrial energy metabolism. We interpret these data as evidence that MIRO1 is critical for maintaining [Ca^2+^]_mito_, which is essential for increased energy production during the cell cycle.

Reconstitution of MIRO1^−/−^ fibroblasts with MIRO1 mutants produced informative outcomes. Reconstitution with MIRO1 dnC normalized MERCS formation, mitochondrial ATP levels, and cell proliferation; MIRO1 KK or MIRO1 ΔTM did not or only partly restored these parameters. These data align with previous findings that the EF-hand domains of MIRO1 are critical for its full functionality [30,47,48], and that mitochondrial arrest in regions of high Ca^2+^ concentration is driven by the Ca^2+^-sensitive EF hands [37,49]. Indeed, without the intracellular mitochondrial arrest by Ca^2+^ coordinated by the EF-hand domains or attachment to the outer mitochondrial membrane by the transmembrane domain, proper alignment at MERCS and transfer of Ca^2+^ is impaired. In addition, although MIRO1′s GTPases are evolutionarily conserved, the functional significance of the C-terminal GTPase is less robust compared with the N-terminal GTPase, as mutants in this domain result in a phenotype similar to WT MIRO1 [50,51]. This report corroborates those data, as MIRO1 dnC expression restored MERCS formation, ATP levels, and proliferation in a manner comparable to MIRO1 WT.

This study has several limitations. Among the limitations in this study is the interpretation of proliferation in the in vivo mouse model of wound healing. The murine excisional wound healing model progresses through overlapping phases: inflammation, proliferation, and remodeling [52]. In our studies, we attributed the reduction in wound size to fibroblast proliferation, which occurs between days 3 and 14 [52]. However, this interpretation does not account for other factors involved in wound healing, such as neutrophil infiltration, macrophage migration, and collagen deposition. In addition, in fibroblasts explanted from the in vivo model of MIRO1 deletion, we detected only a 50% reduction in MIRO1 expression after tamoxifen injections. The presence of multiple cell types in skin specifically at earlier passages after isolation is likely a contributing factor. Another limitation in this study is the quantification of protein expression in MAM fractions from cultured cells. In other studies, the purity of the MAM fraction was determined based on the enrichment of proteins, such as VDAC1 and FACL4 [28]. In this study, FACL4 enrichment significantly increased under growth conditions. Thus, MAM proteins were normalized to VDAC1 levels, which remained relatively unchanged despite its action in Ca^2+^ transfer at the contact site. Lastly, this MERCS study focused specifically on MIRO1 and the IP3R-GRP75-VDAC1 contact site. Other MERCS involved in Ca^2+^ transfer include Sigma receptor 1 (SIG1R) and BiP, as well as MFN2-MFN1 [53,54]. MFN2 deletion impaired mitochondrial Ca^2+^ uptake in skeletal muscle and rat aortic VSMCs [55,56].

In summary, these data show that MERCS formation is a dynamic process that is coordinated with the cell cycle, and that changes in MERC density and MAM composition strongly affect mitochondrial Ca^2+^ uptake and ATP production. Thus, mitochondria–ER interactions are a critical requirement for cell cycle progression, and MIRO1 coordinates MERCS formation with the cell cycle and its downstream events.

## Figures and Tables

**Figure 1 cells-14-00482-f001:**
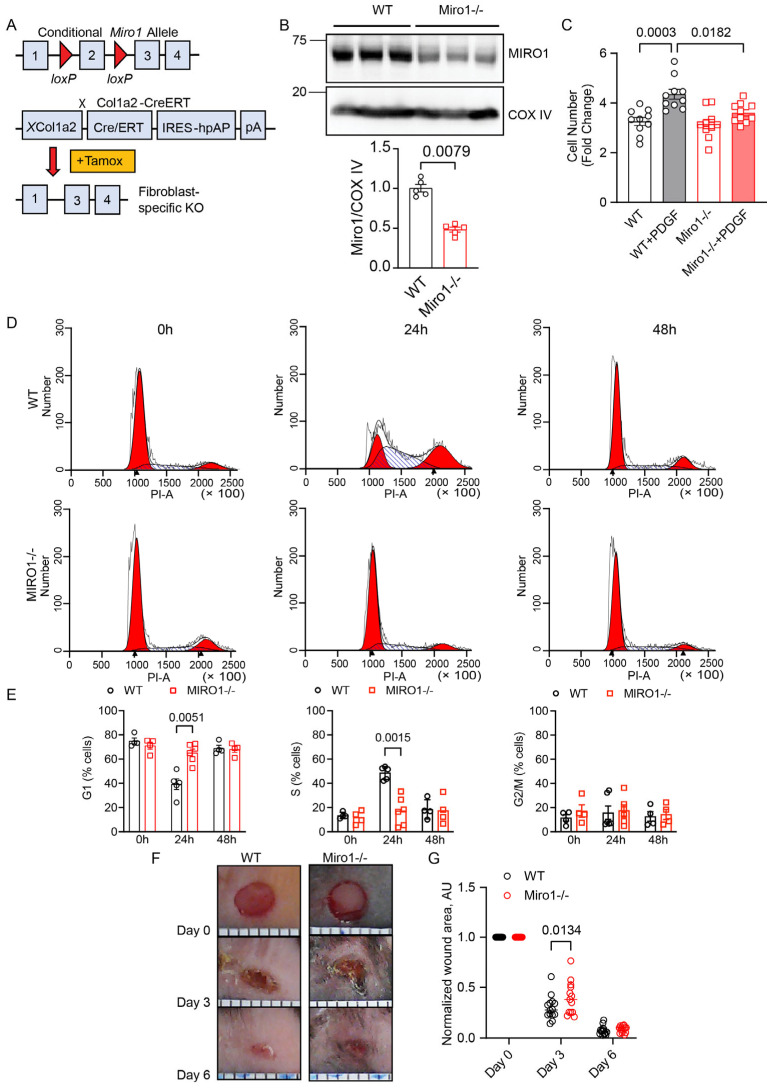
MIRO1 is required for proliferation in vitro and wound healing in vivo. (**A**) Schematic depicting the genetic strategy used to generate fibroblast-specific MIRO1^−/−^ mice. Miro1^fl/fl^ mice were crossed with mice expressing tamoxifen-inducible, fibroblast-specific Cre recombinase, Col1a2CreERT. Tamoxifen was administered (80 mg/kg/day) for a total of 10 days to induce fibroblast-restricted Cre expression. (**B**) Representative immunoblot for MIRO1 in mitochondrial fractions of lysates from the skin of WT and Miro1^−/−^ mice after wound closure. The quantification of MIRO1 protein is adjusted to COX IV; n = 5 mice per group. (**C**) Cell counts of skin fibroblasts explanted from WT and Miro1^−/−^ mice incubated in media containing 10% FBS with and without PDGF for 72 h (20 ng/mL); n = 10 independent experiments. (**D**) Representative FACS analysis for DNA content in synchronized/growth-arrested WT and MIRO1^−/−^ skin fibroblasts at 0 h and after release from arrest with 10% FBS for 24 h and 48 h. (**E**) Cell cycle phase distribution (% of cells) of skin fibroblasts in the G1, S, and G2/M phases; n = 4–6 independent experiments. (**F**) Representative images of wounds after intrascapular skin punch at days 0 (immediately after punch), 3, and 6 in WT and Miro1^-/-^ mice. The scale depicted below the images represents 1 mm. (**G**) Quantification of wound areas. Data were normalized to the wound area at day 0; n = 14 mice per genotype. Data are shown as the mean ± SEM. Analyses were performed using the Mann–Whitney test (**B**), one-way ANOVA (**C**), two-way ANOVA (or mixed model) (**E**), or two-way ANOVA (**G**).

**Figure 2 cells-14-00482-f002:**
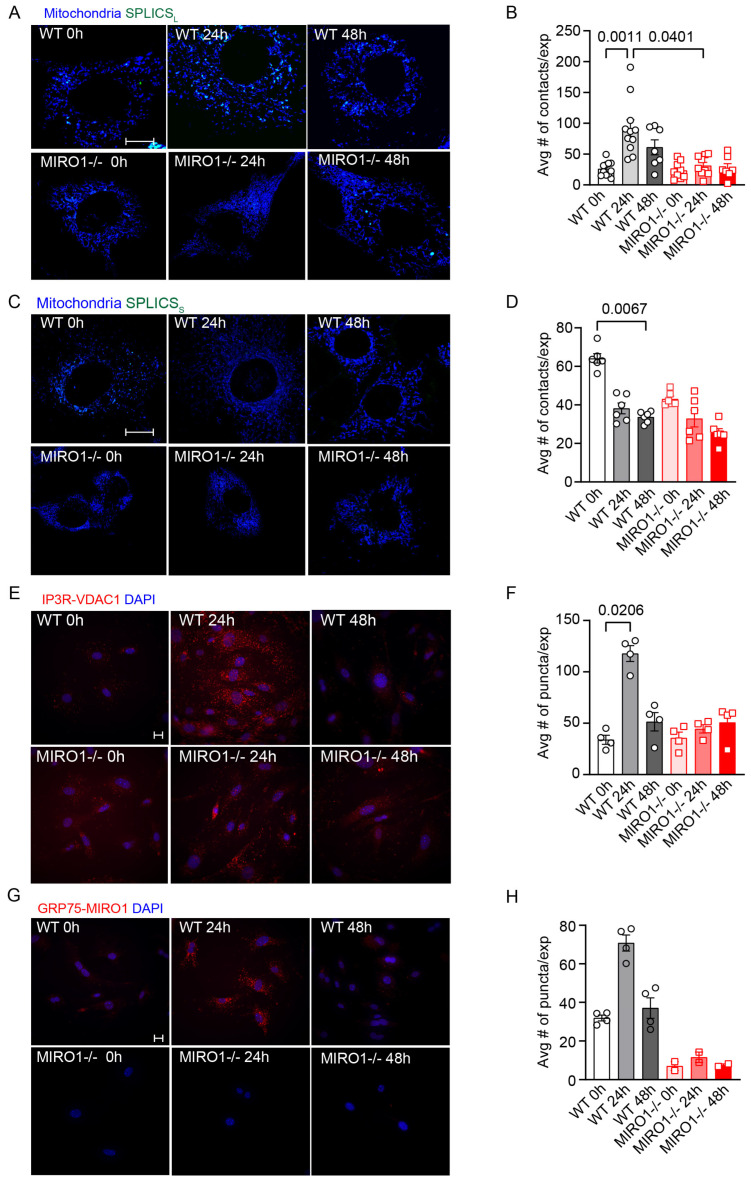
MIRO1 regulates the number of mitochondria–ER contacts during the cell cycle. (**A**) Representative images of VSMCs expressing a split-GFP-based contact-site sensor for wide juxtaposition (40–50 nm) between the ER and mitochondria (SPLICS_L_, green) colocalized with mitochondria (MitoTracker, blue) in WT and MIRO1^−/−^ cells. Scale bar = 20 µm, ×63. (**B**) Quantification of SPLICS_L_ in (**A**); n = 7–11 independent experiments. (**C**) Representative images of VSMCs expressing a split-GFP-based contact-site sensor for narrow juxtaposition (8–10 nm) between the ER and mitochondria (SPLICS_S_; green) colocalized with mitochondria (MitoTracker, blue) in WT and MIRO1^−/−^ cells. Scale bar = 20 µm, ×63. (**D**) Quantification of SPLICS_S_ in (**C**); n = 6 independent experiments. (**E**) Representative images of the in situ proximity ligation assay (PLA) between IP3R and VDAC1. PLA products are shown in red and the nucleus in blue (DAPI). Scale bar = 20 µm, ×40. (**F**) Quantification of the images in (**E**); n = 4 independent experiments. (**G**) Representative images of the in situ proximity ligation assay (PLA) between GRP75 and MIRO1. PLA products are shown in red and the nucleus in blue (DAPI). Scale bar = 20 µm, ×40. (**H**) Quantification of the images in (**G**); n = 2–4 independent experiments. Data are shown as the mean ± SEM. Analyzed using the Kruskal–Wallis test.

**Figure 3 cells-14-00482-f003:**
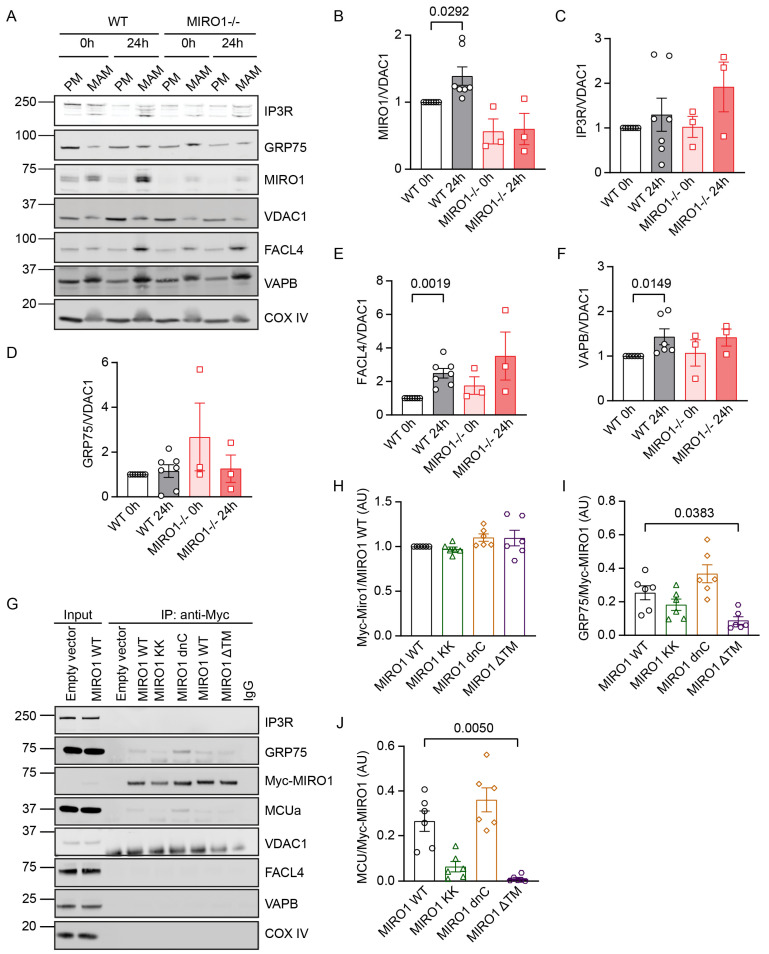
MIRO1 resides at MAM interfaces and interacts with Ca^2+^-transfer MERCS proteins. (**A**) Representative immunoblots for MERCS proteins in fractions of purified mitochondria (PM) and of mitochondria-associated membranes (MAMs) isolated from WT and MIRO1^−/−^ skin fibroblasts following synchronization in serum-free medium (0 h) and at 24 h after release from growth arrest in medium containing 10% FBS. PM: purified mitochondria, MAM: mitochondria-associated membrane. Markers for MAMs and ER (FACL4) and mitochondria (cytochrome c oxidase (COX IV)) were also examined. VDAC1 was used as a loading control. (**B**–**F**) Quantification of the immunoblot experiments as in (**A**). (**B**) MIRO1, (**C**) IP3R, (**D***)* GRP75, (**E**) FACL4, and (**F**) VAPB levels, adjusted to VDAC1; n = 3–7 independent experiments. (**G**) Coimmunoprecipitation (co-IP) analysis of MIRO1 WT, MIRO1 KK, MIRO1 dnC, and MIRO1 ΔTM with MERCS proteins. c-Myc-tagged MIRO1 constructs were expressed in HEK cells for 24 h, and cell lysis and pull-down assays were performed. (**H**–**J**) Quantification of the co-IP experiments shown in (**G**). (**H**) MIRO1 expression in MIRO1 KK, MIRO1 dnC, and MIRO1 ΔTM adjusted to MIRO1 WT. (**I**) GRP75 and (**J**) MCU levels, adjusted for immunoprecipitated c-Myc-tagged MIRO1; n = 6 independent experiments. Data are shown as the mean ± SEM. Analyses were performed using the Kruskal–Wallis test.

**Figure 4 cells-14-00482-f004:**
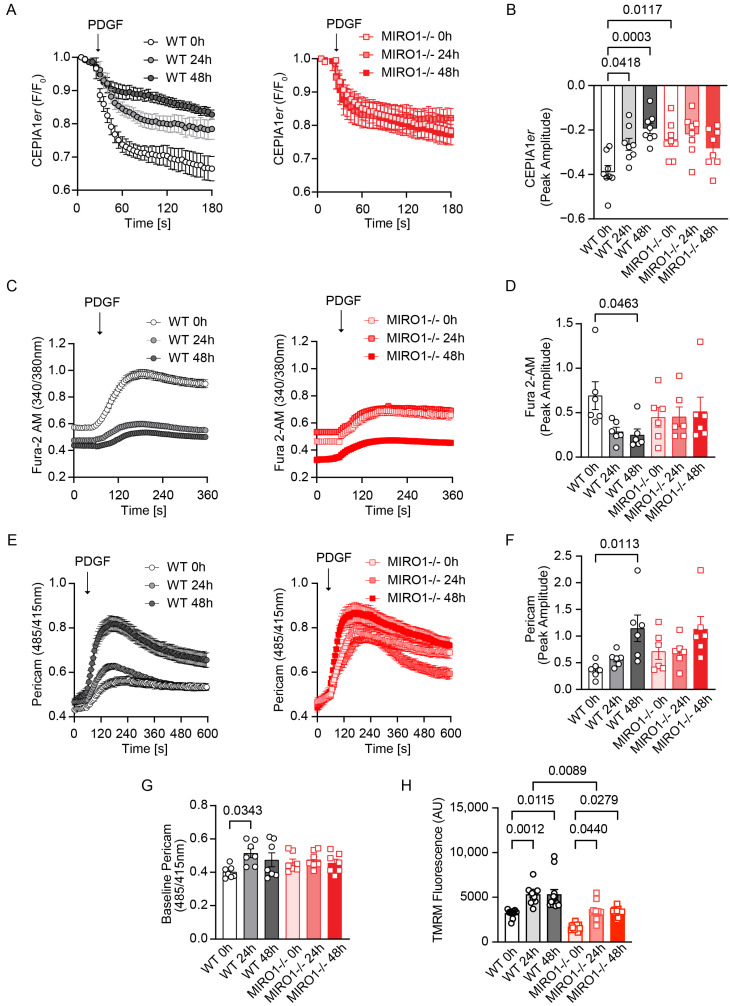
MIRO1 regulates changes in subcellular Ca^2+^ distribution during the cell cycle. (**A**) PDGF-induced ER Ca^2+^ release as assessed with CEPIA1*er* in synchronized/growth-arrested WT and MIRO1^-/-^ VSMCs at 0 h and after release from arrest with 10% FBS for 24 h and 48 h. Arrows indicate the addition of PDGF (20 ng/mL). (**B**) Quantification of the peak amplitude of CEPIA1*er* recordings shown in (**A**); n = 8 independent experiments. (**C**) PDGF-induced cytosolic Ca^2+^ transients as assessed with Fura 2-AM in synchronized/growth-arrested WT and MIRO1^−/−^ VSMCs at 0 h and after release from arrest with 10% FBS for 24 h and 48 h. Arrows indicate the addition of PDGF (20 ng/mL). (**D**) Quantification of the peak amplitude of Fura 2-AM recordings shown in (**C**); n = 6 independent experiments. (**E**) PDGF-induced mitochondrial Ca^2+^ uptake as assessed with mtPericam in synchronized/growth-arrested WT and MIRO1^-/-^ VSMCs at 0 h and after release from arrest with 10% FBS for 24 h and 48 h. Arrows indicate the addition of PDGF (20 ng/mL). (**F**) Quantification of the peak amplitude of mtPericam recordings shown in (**E**); n = 6 independent experiments. (**G**) Quantification of baseline mitochondrial [Ca^2+^] as assessed with mtPericam in synchronized/growth-arrested WT and MIRO1^−/−^ VSMCs at 0 h and after release from arrest with 10% FBS for 24 h and 48 h; n = 7 independent experiments. (**H**) Quantification of the mitochondrial membrane potential as assessed by tetramethylrhodamine methyl ester (TMRM) fluorescence in synchronized/growth-arrested WT and MIRO1^−/−^ VSMCs at 0 h and after release from arrest with 10% FBS for 24 h and 48 h; n = 8 independent experiments. Data are shown as the mean ± SEM. Analyses were performed using one-way ANOVA (**B**) and Kruskal–Wallis (**D**,**F**–**H**) tests.

**Figure 5 cells-14-00482-f005:**
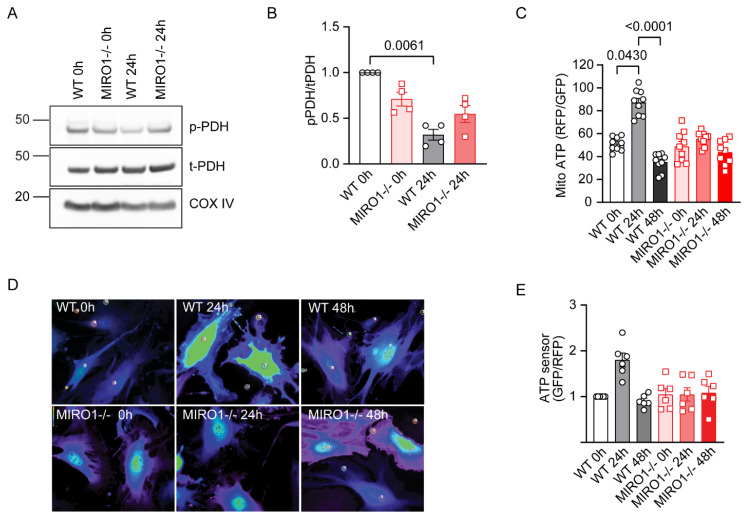
Loss of MIRO1 attenuates mitochondrial and cytosolic ATP levels. (**A**) Representative immunoblots of phosphorylated (inactive) pyruvate dehydrogenase (p-PDH) and total pyruvate dehydrogenase (t-PDH) in whole-cell lysates of WT and MIRO1^−/−^ VSMCs at 0 h and after release from arrest with 10% FBS for 24 h. (**B**) Quantification of p-PDH (α1-ser293), adjusted to t-PDH. COX IV was used as a loading control; n = 4 independent experiments. (**C**) Quantification of mitochondrial ATP levels in synchronized/growth-arrested WT and MIRO1^−/−^ VSMCs at 0 h and after release from arrest with 10% FBS for 24 h and 48 h; n = 9 independent experiments. (**D**) Representative images of WT and MIRO1^-/-^ VSMCs transduced with adenovirus expressing the fluorescent ATP-sensitive protein pm-iATPSnFR1.0 synchronized/growth-arrested at 0 h and after release from arrest with 10% FBS for 24 h and 48 h. (**E**) Quantification of the cytosolic ATP levels shown in (**D**); n = 6 independent experiments. Data are shown as the mean ± SEM. Analyses were performed using the Friedman (**B**,**E**) and Kruskal–Wallis (**C**) tests.

**Figure 6 cells-14-00482-f006:**
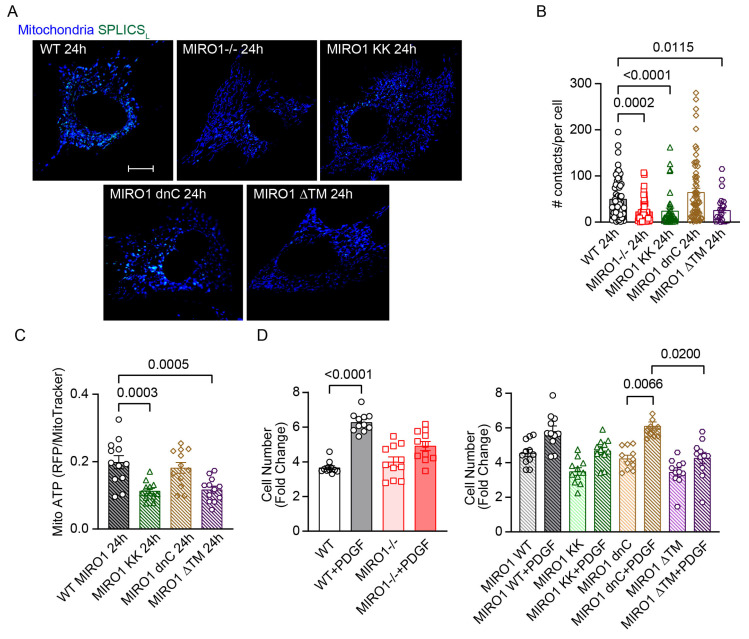
MIRO1 EF hands and transmembrane domain is required for MERCS formation, increased ATPlevels and cell proliferation in skin fibroblasts. (**A**) Representative images of WT skin fibroblasts and MIRO1^−/−^ skin fibroblasts expressing MIRO1 KK, MIRO1 dnC, or MIRO1 ΔTM and a split-GFP-based contact-site sensor for wide juxtaposition (40–50 nm) between the ER and mitochondria (SPLICS_L_; green) colocalized with mitochondria (MitoTracker; blue) after release from arrest with 10% FBS for 24 h. Scale bar = 20 µm, ×63. (**B**) Quantification of the SPLICS_L_ shown in (**A**); n = 30 to 65 cells for each group from 5 independent experiments. (**C**) Quantification of mitochondrial ATP levels at 24 h in WT skin fibroblasts and MIRO1^−/−^ skin fibroblasts expressing MIRO1 KK, MIRO1 dnC, or MIRO1 ΔTM after release from growth arrest with 10% FBS; n = 12 independent experiments. (**D**) Cell counts of WT skin fibroblasts and MIRO1^−/−^ skin fibroblasts incubated in media containing 10% FBS with and without PDGF for 72 h (20 ng/mL); n = 10 independent experiments. (**E**) Cell counts of MIRO1^−/−^ skin fibroblasts expressing MIRO1 WT, MIRO1 KK, MIRO1 dnC, or MIRO1 ΔTM incubated in media containing 10% FBS with and without PDGF for 72 h (20 ng/mL); n = 10 independent experiments. Data are shown as the mean ± SEM. Analyses were performed using Kruskal–Wallis (**B**,**D**,**E**) and one-way ANOVA (**C**) tests.

## Data Availability

The additional data are contained within the Appendix A. Further inquiries regarding the raw data can be directed to the corresponding author.

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
