# Peer review of "MIRO1 Is Required for Dynamic Increases in Mitochondria-ER Contact Sites and Mitochondrial ATP During the Cell Cycle"

_cells, 2025, doi:10.3390/cells14070482_

Round 1
Reviewer 1 Report
Comments and Suggestions for Authors
In this manuscript, the authors report that MIRO1 can control an increase in the number of mitochondrial ER contact sites (MERCS) through its interactions with GRP75 in the G1/S phase during the cell cycle. This subsequently causes increased ER Ca2+ release and mitochondrial Ca2+ uptake, which leads to increased mitochondrial ATP production, cell proliferation, and wound healing. For this phenomenon, the Ca2+-sensitive EF-hands and transmembrane domain of MIRO1 are required. The topic of this study is of interest, but there are a number of concerns that undermine confidence in these conclusions.
1. Figure 4A-D: In control cells (WT), PDGF-induced ER Ca2+ contents (A-B) and cytosolic Ca2+ levels (C-D) were highest at 0 h and significantly decreased by 48h, but those in MIRO1 knockdown cells (referred to as MIRO1-/-) were reduced at 0 h compared to controls and remained unchanged at later time points. The authors interpreted these observations that MIRO1 deficiency altered ER Ca2+ release during the cell cycle. Since the elevated cytosolic Ca2+ levels are triggered by multiple mechanisms (i.e., ER Ca2+ release, ER Ca2+ uptake, Ca2+ extraction to the plasma membrane, and Ca2+ uptake to the mitochondria), the authors need to dissect, verify, and discuss the relevant mechanisms precisely. To support the authors’ interpretation and conclusions, direct measurements of PDGF-induced IP3R activity and IP3 production levels in WT and MIRO1-/- cells are needed.
2. Figure 4E-F: In WT cells, PDGF-induced mitochondrial Ca2+ uptake increased significantly by 48 h, but in MIRO1-/- cells, mitochondrial Ca2+ uptake did not significantly change during the cell cycle. The authors propose multiple mechanisms as a cause of these changes in mitochondrial Ca2+ uptake; 1) ER Ca2+ release, 2) MERCS structures (e.g., number, distance, and size), and 3) mitochondrial membrane potential. What is a major cause of this abolished increase in PDGF-induced mitochondrial Ca2+ uptake in MIRO1 -/- cells during the cell cycle? Please dissect, compare, and discuss the relative contributions of MIRO1 to each potential mechanism. To precisely demonstrate the role of MIRO1-mediated MERCS on mitochondrial Ca2+ uptake, additional experiments likely need to normalize ER Ca2+ release and then measure and compare mitochondrial Ca2+ uptake in response to PDGF in WT and MIRO1-/- cells.
3. Figure 4E-F: No summary data of the average time-to-peak of mitochondrial Ca2+ uptake is shown. Please show it and discuss whether or not the changes in the peak amplitude are due to changes in average time-to-peak.
4. Figures 2, 4, and 5: These data sets are generated from vascular smooth muscle cells (VSMCs) isolated from MIRO1 fl/fl mice following transduction with adenoviruses expressing Cre recombinase or control according to the information in the Method section. Based on the author’s group publication (Quian L et al., bioRxiv. 2024. doi: 10.1101/2024.08.13.607854), the MIRO1-mediated cell cycle progression and proliferation mechanisms in VSMCs seem to be quite distinct compared to those in skin fibroblasts. Please justify the use of VSCMs for this study.
5. Figure 1D-E: It is unclear if the data sets are generated from skin fibroblasts isolated from WT and fibroblast-specific MIRO1-knockout mice or MIRO1 fl/fl mice following transduction with adenoviruses expressing Cre recombinase and control. Please clarify and include some details of the experimental conditions in the Figure legends.
6. No detailed information on MIRO1 mutants (i.e., mutation sites, deletion sites, and so on) is written in the manuscript.
7. Figure 6C: The bar graph labels do not match the information in the legend. Please confirm this.
Reviewer 2 Report
Comments and Suggestions for Authors
In this paper titled ‘MIRO1 is require for dynamic increases in mitochondrial ER contact sites and mitochondrial ATP during cell cycle’, authors study how MIRO1 regulates the dynamics of mitochondrial ER contact sites (MERCs) and thereby affects proliferation and mitochondrial ATP during cell cycle. They designed experiments to evaluate proliferation and MERCs changes during cell cycle using FBS starvation and growth release-based assays. Using this system, they studied the dynamic changes in MERC proteins and interaction in MIRO1-deficient cell lines. Overall, they have found that MIRO1 deficiency blocked G1/S progression and cell cycle-dependent formation of MERCS as well as calcium alterations. While full length MIRO1 could restore these defects, the MIRO1 mutant lacking calcium EF hand or the transmembrane domain could not rescue cell proliferation and MERC formation, showing dierct role of MIRO1 in these processes.
Although the paper is interesting and provides new insights about the role of MIRO1 and MERCs in cell cycle progression, it appears that several control experiments-especially for fluorescence assays-are missing, making it prone to misinterpretations. Adequate control experiments are critical to correctly interpret these results and make meaningful conclusions. This also applies to some of the quantification of western blot analysis. I would strongly suggest to address these comments before the paper is accepted. These experiments are crucial for supporting the claims made by the authors.
Major comments:
1. Use of SPLICSL and SPLICSS: While these probes are suitable, the authors must include positive and negative conditions to determine the baseline assay range and ensure that the results are not due to misinterpretations. For example, how do authors know that the decrease in GFP spots in MIRO1 KO cells is not due to reduced expression or malfunctioning of the construct. A positive condition in these cell lines should be included to validate the assay results. How are the puncta quantified? In my opinion, a complementary assay like electron microscopy or other fluorescence technique will be useful to verify this change in MERCs rather than relying on only one assay.
2. The images that are provided in the paper for PLA assay are of very low resolution and quality. There are no clear visible puncta, and the resolution is insufficient to reliably determine the number of puncta. Author should clarify how the number of puncta were determined from these images. It would be worthwhile to also provide a zoom image to clearly mark the puncta. Here also I suggest to include complementary assay to verify the results.
3. In Figure 3G co-ip experiment, the interaction between MICRO1 and GRP75 appear weak and questionable. The input lane is oversaturated and still the signal in the elution fraction is barely visible. Why are these interactions so weak, is it that very tiny amount of GRP75 interacts with MIRO1 or that the co-ip conditions are not optimized enough to capture the MERC complex? Additionally, the quantification of such a blot in Fig 3I seems unreliable due to vast differences in signal intensities. Even if the results could be reproducible, the lack of linearity in the quantification range makes the interpretations prone to error.
Minor comments:
Typos- The typos (is) in title section 3.7
In Figure 3I- y axis wrongly labelled as MIRO11.
Reviewer 3 Report
Comments and Suggestions for Authors
The article by Endoni et al is devoted to the role of MIRO1 in the dynamics of the formation of mitochondria ER contact sites during the cell cycle. Wild-type and MIRO1-/- fibroblasts and vascular smooth muscle cells were evaluated for proliferation, cell cycle progression, MERCS number, distance, and protein composition throughout the cell cycle. The authors also showed changes in the energy metabolism of cells with reduced Miro1 expression. The authors make the key conclusion that MIRO1 controls an increase in the number of MERCS during cell-cycle progression and increases mitochondrial [Ca2+], driving metabolic activity and proliferation through its EF hands. The article is well structured and illustrated. However, there are some comments to the work.
Comments
1. Why did the authors use fibroblasts and vascular smooth muscle cells in their study? In the materials and methods section, the authors described that they used male and female mice in their work. Why did the authors take cells from mice of different sexes?
2. In Fig. 3, the authors used VDAC1 as a loading control. However, Fig. 3A shows that the level of VDAC1 changes. Therefore, it is incorrect to use such a loading protein. I would like to recommend the authors to use COX4 as a loading control.
3. The authors showed that during the cell cycle in wild-type cells there is a change in MERCS, as well as Ca2+ homeostasis, mitochondrial membrane potential and ATP (0h, 24h, 48h). The authors' comments on the fact that the ATP level and the mitochondrial membrane potential increase simultaneously in the 24h phase are interesting. It can be assumed that in this phase there is a slowdown in cellular metabolism, when ATP is not consumed and, consequently, the membrane potential does not decrease. How can this be linked to the increase in Ca2+ levels? Could this indicate a cessation of cellular metabolism? In the WT 48h phase, the ATP level drops, but the membrane potential does not change. How can these data be interpreted? At the same time, in MIRO1-/- cells, the G0/G1 phase pattern is observed in all phases. What, in the authors' opinion, is the cell cycle arrest in these cells associated with? Only with the disruption of the bioenergetic function or with possible other processes.
Round 2
Reviewer 1 Report
Comments and Suggestions for Authors
The authors adequately addressed the reviewer's concerns.
Reviewer 2 Report
Comments and Suggestions for Authors
The authors have incorporated the changes suggested to them, which have significantly improved the manuscript. They have now provided the controls for fluorescence assays and other critical control experiments. The addition of TEM data further supports the other assays, serving as a verification of the results. Overall, I believe the paper is suitable for publication in this form.
Reviewer 3 Report
Comments and Suggestions for Authors
The manuscript is appropriately revised and it can be published in the current state.